# Quantifying and Learning Disentangled Representations with Limited Supervision

## Abstract

Learning low-dimensional representations that disentangle the underlying factors of variation in data has been posited as an important step towards interpretable machine learning with good generalization. To address the fact that there is no consensus on what disentanglement entails, Higgins et al. (2018) propose a formal definition for Linear Symmetry-Based Disentanglement, or LSBD, arguing that underlying real-world transformations give exploitable structure to data.

Although several works focus on learning LSBD representations, such methods require supervision on the underlying transformations for the entire dataset, and cannot deal with unlabeled data. Moreover, none of these works provide a metric to quantify LSBD.

We propose a metric to quantify LSBD representations that is easy to compute under certain well-defined assumptions. Furthermore, we present a method that can leverage unlabeled data, such that LSBD representations can be learned with limited supervision on transformations. Using our LSBD metric, our results show that limited supervision is indeed sufficient to learn LSBD representations.

## 1 Introduction

Disentangled representation learning aims to create low-dimensional representations of data that separate the underlying explanatory factors of variation in data. These representations provide an interpretable (Sarhan et al., 2019) and useful tool for various purposes, such as noise removal (Lopez et al., 2018), continuous learning (Achille et al., 2018), and visual reasoning (van Steenkiste et al., 2019).

However, there is no consensus about the exact properties that characterize a disentangled representation. Higgins et al. (2018) provide a formal definition for Symmetry-Based Disentangled (SBD) and Linearly SBD (LSBD) data representations, building upon the idea that representations should reflect the underlying structure of the data. In particular, they argue that variability in the data comes from transformations in the real world from which the data is observed. Having a formal definition of disentanglement can serve as a paradigm for the evaluation of disentangled representations.

Although several methods have been proposed to learn SBD or LSBD representations, none of them provide a clear metric for quantifying the level of disentanglement in these representations. Quessard et al. (2020) introduce a loss term that measures the complexity of the transformations acting on their learned representations based on the number of parameters needed, but this term does not directly characterize disentanglement. Caselles-Dupré et al. (2019) only evaluate the performance of their learned representations when used in a particular downstream task.

Moreover, existing methods require information about the transformation relationships among data points for the entire training dataset. This information is used to produce models that enforce the properties of SBD or LSBD representations and can be considered as a form of supervision. For example, this supervision can consist of the parameters of the transformation that connects a pair of data points, such as a rotation angle. Obtaining this supervision on the transformations for a dataset can be an expensive task that requires expert knowledge.

In this work we focus on characterizing and quantifying LSBD and developing a method capable of obtaining LSBD representations by using a limited amount of supervision on the transformation properties of a dataset. The main contributions of this paper are:

1. An easy-to-compute metric to quantify LSBD given certain assumptions (see Section 4), which acts as an upper bound to a more general metric (derived in Appendix D).

2. A partially supervised method to obtain LSBD representations that, during training, can also use data without supervision on the transformation relationships.

## 2 RELATED WORK

The concept of disentanglement comes from the intuition that data can be described in terms of a set of independent explanatory factors that constitute the variability of the data. In probabilistic modeling, these factors are interpreted as random independent unobserved latent variables that condition the data generation process (Kulkarni et al., 2015; Higgins et al., 2016; Chen et al., 2016; 2018). Nevertheless, there is no consensus about the exact properties that disentangled representations should have. Evidence of this is the wide range of metrics used to characterize such representations (Locatello et al., 2018). However, a reasonable expectation is that such representations should capture and separate the variations and properties of the data.

Recent work has turned the attention to capturing not only the independent explanatory factors of a dataset, but also the transformations that those factors undergo in data generation. Examples of methods that attempt to produce representations with similar transformation properties to those of the explanatory factors of data are (Cohen & Welling, 2015; Worrall et al., 2017; Sosnovik et al., 2019).

Transformations of the real world determine the variability of data and its structure. These so-called symmetry transformations have long been studied in Physics (Gross, 1996) and have been formalized with group theory. Such transformations often affect only a subset of the properties that describe the real world and leave the rest invariant.

The connection between disentanglement and symmetries has recently been formalized by Higgins et al. (2018) into the definitions of Symmetry-Based Disentangled (SBD) and Linearly SBD (LSBD) representations. Quessard et al. (2020); Caselles-Dupré et al. (2019) propose methods to obtain such SBD and/or LSBD representations, but their methods require supervision on the transformation relationships among datapoints for the entire training dataset. Moreover, there is no clear metric for the level of disentanglement in their methods.

## 3 SYMMETRY-BASED DISENTANGLEMENT

Higgins et al. (2018) provide a formal description of disentanglement that connects the symmetry transformations affecting the real world (from which data is generated) to the internal representations of a model. The definitions are grounded in concepts from group theory, for a more detailed description of these concepts please refer to Appendix A.

These definitions assume the following setting. $W$ is the set of possible world states, with underlying symmetry transformations that are described by a group $G$ and its action $\cdot : G \times W \to W$ on $W$. In particular, $G$ can be decomposed as the direct product of $K$ groups $G = G_1 \times \ldots \times G_K$. Data is obtained via an *observation* function $b : W \to X$ that maps world states to observations in a *data space* $X$. A model's internal representation of data is modeled with the *inference* function $h : X \to Z$ that maps data to the *representation space* $Z$. Together, the observation and the inference constitute the model's internal representation of the real world $f : W \to Z$ with $f(w) = h \circ b(w)$.

The definitions for Symmetry-Based Disentangled (SBD) and Linearly SBD (LSBD) representations formalize the requirement that a model's internal representation $f : W \to Z$ should reflect and disentangle the transformation properties of the real world. The definition of SBD can be found in Appendix B. In particular, our work focuses on LSBD representations in which the transformation properties of the model's internal representations should be linear. The exact definition is as follows:

**Linearly Symmetry-Based Disentangled (LSBD) Representations** A model's internal representation $f : W \to Z$, where $Z$ is a vector space, is LSBD with respect to the group decomposition $G = G_1 \times \ldots \times G_K$ if

- there is a decomposition of the representation space $Z = Z_1 \oplus \ldots \oplus Z_K$ into $K$ vector subspaces,
- there are group representations for each subgroup in the corresponding vector subspace $\rho_k : G_k \to GL(Z_k), k \in \{1, \ldots, K\}$
- the group representation $\rho : G \to GL(Z)$ for $G$ in $Z$ acts on $Z$ as

$$\rho(g) \cdot z = (\rho_1(g_1) \cdot z_1, \ldots, \rho_K(g_K) \cdot z_K), \tag{1}$$

for $g = (g_1, \ldots, g_K) \in G$ and $z = (z_1, \ldots, z_K) \in Z$ with $g_k \in G_k$ and $z_k \in Z_k$.
- the map $f$ is *equivariant* with respect to the actions of $G$ on $W$ and $Z$, i.e. , for all $w \in W$ and $g \in G$ it holds that $f(g \cdot w) = \rho(g) \cdot f(w)$.

**Symmetry-based disentanglement with respect to the data space** $X$     We do not directly observe $W$, only data observations in $X$, thus it is most practical to evaluate (L)SBD with respect to the inference map $h : X \to Z$. However, the definitions for SBD and LSBD representations are given with respect to the world states representation map $f : W \to Z$. If the action of $G$ on $W$ is *regular* and $b$ is *injective* though, a representation $f$ is (L)SBD if the inference $h$ restricted to all possible observations is also (L)SBD. In this case, we can evaluate the same conditions for (L)SBD only for the function $h$. For a more detailed description on these conditions consult Appendix C.

## 4 ASSUMPTIONS

In this paper we assume a setting where an underlying group structure and its action on the data space and representation space are provided as expert knowledge, which allows us to provide a metric that quantifies Linear Symmetry-Based Disentanglement (LSBD) and a method to achieve it. Although these assumptions limit the settings in which our disentanglement metric and method can be used, we believe it is valuable because (1) it provides a good basis for future research into more general metrics and methods by making clear which assumptions should be relaxed, (2) it gives good insights into the effect of the level of supervision on transformations (as shown by our results in Section 7), and (3) there are many practical scenarios where having this type of expert knowledge is a reasonable assumption.

More concretely, we make the following assumptions about the available information about the data and its transformations:

1. Let $W$ be the set of real world states. Let $X$ be the data space, where observations are obtained. Let $Z$ be the representation space where we wish to model this data; $Z$ is a vector space with given basis.

2. We have a Lie group $G$ and a *regular* action of $G$ on $W$ that describes the underlying symmetries of $W$. The group decomposes as the direct product of groups $G = G_1 \times \ldots \times G_K$.

3. The observation function $b : W \to X$ is *injective*. Therefore, we can define a group action for $G$ on the image of $b$, i.e. $\cdot : G \times b(W) \to b(W)$ as $g \cdot x = b(g \cdot b^{-1}(x))$ (see Appendix C for more details). This action is also *regular*. Practically, this implies that each observation is a transformed version of another observation, and this transformation is unique.

4. We have an action $\cdot : G \times Z \to Z$ that is linear and has a known representation $\rho : G \to GL(Z)$, i.e. for any $g \in G$ we can determine $\rho(g)$.

5. The representation $\rho : G \to GL(Z)$ is *linearly disentangled*, i.e. there is a known decomposition $Z = Z_1 \oplus \ldots \oplus Z_k$ such that $\rho(g) \cdot z = (\rho_1(g_1) \cdot z_1, \ldots, \rho_K(g_K) \cdot z_K)$, with $g_k \in G_k, z_k \in Z_k$ for $k = 1, \ldots, K$.

6. There exists an embedded submanifold $Z_G \subseteq Z$ such that the action of $G$ on $Z$ restricted to $Z_G$ is *regular*, and $Z_G$ is invariant under the action. Only $Z_G$ will be used as the codomain of the inference map, $h : X \to Z_G$.

**Example: the special orthogonal group** $SO(2)$ **of 2D rotations**     Let's look at an example that satisfies the assumptions above and can reasonably occur in practice. Consider a windmill that can be rotated around a fixed vertical axis, with blades that can rotate independently of the windmill's orientation. The dataset consists of observations (i.e. 2D images) of this windmill, where the only

variability is in the rotations of the windmill and of its blades, all other aspects (e.g. camera angle, lighting conditions, object position, etc.) are fixed. Thus, the variability in this dataset can be described as two independent 2D rotations, see Figure 1.

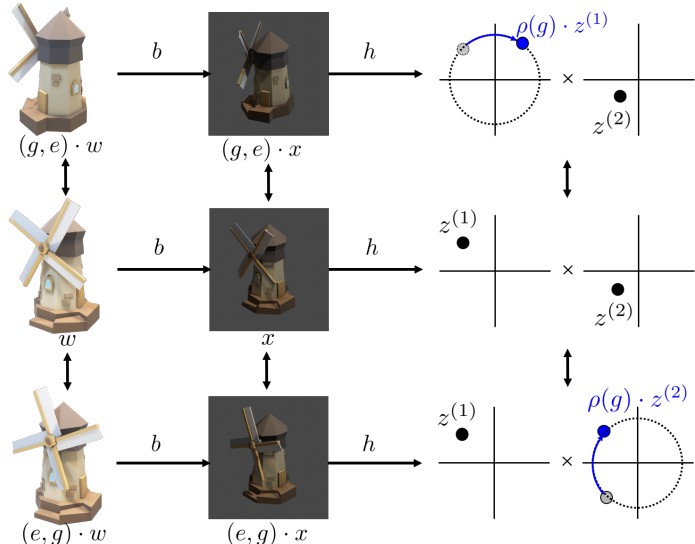

Figure 1: Illustration of an LSBD representation with respect to the decomposition $G = SO(2) \times SO(2)$ for the windmill example, 3D model by (Poly). The action of $G$ on the real world is decomposed as two independent transformations: the rotation of the windmill with respect to a vertical axis and the rotation of the blades. The inference function $h : X \rightarrow Z$ maps images into $Z = \mathbb{R}^2 \oplus \mathbb{R}^2$. On $Z$, the action of $G$ is the independent rotation of each vector subspace. The action of $G$ on $Z_G = S^1 \times S^1 \subseteq Z$ is regular.

We can satisfy the assumptions above as follows:

1. The world states $W$ can be described by the angle of the entire windmill and the angle of the blades, these angles are independent of each other. The data space $X$ is the space of images, i.e. arrays of pixel values. A suitable representation space $Z$ will become apparent soon.

2. It is well-known that 2D rotations can be described by the special orthogonal group $SO(2)$, so the transformations underlying this dataset can be described with the group decomposition $G = G_1 \times G_2 = SO(2) \times SO(2)$. The corresponding action is indeed *regular*; any two world states are connected with one unique transformation (i.e. pair of rotations).[1]

3. If the windmill looks different from each angle, and the blades are always visible, then the observation function $b : W \rightarrow X$ is indeed *injective*.

4. Each subgroup $G_k = SO(2)$ has a representation $\rho_k : SO(2) \rightarrow GL(\mathbb{R}^2)$ as rotation matrices for vectors in $\mathbb{R}^2$, so we can use $Z = Z_1 \oplus Z_2 = \mathbb{R}^2 \oplus \mathbb{R}^2$ as the representation space, and the direct sum $\rho = \rho_1 \oplus \rho_2$ as the group representation.

5. Defined in this way, the representation $\rho$ is *linearly disentangled*.

6. The actions of the subgroups $G_i$ on the subspaces $Z_i$ are not *regular*, but they become regular when restricted to the unit circle or 1-sphere $S^1 = \{z \in \mathbb{R}^2 : \|z\| = 1\}$, which is a submanifold of $Z_i = \mathbb{R}^2$ and invariant under the action. Thus, we can use $Z_G = S^1 \times S^1$ as the codomain of the inference map $h$, to obtain a regular group action of $G$ on $Z_G$.

---

[1]Note that for a windmill with e.g. 4 identical blades, a rotation of $\frac{\pi}{2}$ radians corresponds to the identity transformation, and angles can be defined $\mod \frac{\pi}{2}$.

## 5 Quantifying Linear Symmetry-Based Disentanglement

In this section we provide a metric that is easy to compute and characterizes LSBD under the assumptions of Section 4, by quantifying the equivariance of the inference map $h$ with respect to the group action on $X$ and $Z$. Moreover, in Appendix D we show that this metric is an upper bound to a more general LSBD metric that can in theory be evaluated for any encoder.

**Characterizing LSBD representations**    The assumptions from Section 4 ensure that all criteria of the LSBD definition (see Section 3) are satisfied, except for *equivariance* of the inference map $h : X \to Z_G$ with respect to the action of $G$ on $X$ and $Z$. Thus, to evaluate whether $h$ is disentangled, we only need to check if $h(g \cdot x) = g \cdot h(x)$ holds for all $x \in X$ and $g \in G$.

Since the action of $G$ on $X$ is *regular* (see Section 4), the transformations in a dataset $\mathcal{X} = \{x_n\}_{n=1}^N$ (with $x_n \in X$) can be fully described using only $N-1$ transformations. In particular, it is convenient to describe all other data points in terms of the first one: $\mathcal{X} = \{g_n \cdot x_1\}_{n=1}^N$, where $g_1 = e$ is the group identity. Note that $\{g_n\}_{n=2}^N$ can be used to describe the transformation between any two points $x_i, x_j \in \mathcal{X}$ (assuming access to the inverse transformations as well), since $x_i = g_i \cdot (g_j^{-1} \cdot x_j)$.

Equivariance can therefore be evaluated for $\mathcal{X}$ using only $N-1$ comparisons, by testing whether $h(g_n \cdot x_1) = \rho(g_n) \cdot h(x_1)$ for $n = 2, \ldots, N$. To simplify the notation, we write the data representations as $z_n = h(x_n)$ for $n = 1, \ldots, N$. Some rewriting shows that we can evaluate equivariance by checking whether

$$z_1 = \rho(g_2^{-1}) \cdot z_2 = \ldots = \rho(g_N^{-1}) \cdot z_N. \tag{2}$$

This formulation not only gives an efficient way of checking perfect equivariance (and thus LSBD), but also allows for an efficient way to quantify the divergence from this ideal situation, as we will show next.

**Quantifying LSBD**    Equation (2) shows how to check if learned data representations $\{z_n\}_{n=1}^N$ are LSBD, given the transformations $\{g_n\}_{n=2}^N$, but it also provides a perspective to quantify how well the representations are disentangled if perfect disentanglement is not achieved. The transformed representations $\{\rho(g_n^{-1}) \cdot z_n\}_{n=1}^N$ should ideally all be the same, so their *dispersion* is a reasonable measure to quantify equivariance (where no dispersion means perfect equivariance).

An efficient way to measure dispersion is by computing the sample variance. For a sample $\{a_n\}_{n=1}^N$ in Euclidean space, this is computed as $\frac{1}{N} \sum_{n=1}^N \|a_n - \bar{a}\|$, where $\bar{a} = \frac{1}{N} \sum_{n=1}^N a_n$ is the sample mean. In our work however, representations are modeled in a submanifold $Z_G$ that itself is typically not Euclidean. But by defining a metric $d$ on this manifold, we can generalize the sample mean with the Fréchet mean $\bar{a} = \text{mean}_{n=1}^N(a_n) = \arg\min_{a \in Z_G} \sum_{n=1}^N d^2(a_n, a)$, and then compute the Fréchet variance $\frac{1}{N} \sum_{n=1}^N d^2(a_n, \bar{a})$.

Therefore, we propose the following metric (lower is better) to quantify equivariance, and thus LSBD (under the assumptions of Section 4), for a set of representations $\{z_n\}_{n=1}^N \subseteq Z_G$ with in-between transformations $\{g_n\}_{n=2}^N$:

$$\mathcal{L}_{LSBD} = \frac{1}{N} \sum_{n=1}^N d^2(\rho(g_n^{-1}) \cdot z_n, \bar{z}), \tag{3}$$

$$\text{with } \bar{z} := \text{mean}_{n=1}^N(\rho(g_n^{-1}) \cdot z_n), \tag{4}$$

for some metric $d$ defined on the manifold $Z_G$. In particular, $d$ should be preserved under the group representation $\rho$ for this metric to be sensible.

In the example from Section 4 we saw $Z_G = S^1 \times S^1$. In particular, each circle $S^1 = \{z \in \mathbb{R}^2 : \|z\| = 1\}$ is a submanifold of Euclidean space. We choose Euclidean distance as the metric $d$, since it is well-defined on the manifold and preserved under the group representation $\rho$. Moreover, for any two $z, z' \in S^1$, the squared distance between them can be efficiently computed as

$$d^2(z, z') = \|z - z'\|^2 = 2 \cdot (1 - z^T \cdot z'). \tag{5}$$

To compute the squared distance between points in $S^1 \times S^1$ we can simply sum up the squared distances per submanifold $S^1$. With this metric, the Fréchet mean has a simple solution[2]:

$$\underset{n=1}{\overset{N}{\mathrm{mean}}}(z_n) = \frac{\sum_{n=1}^{N} z_n}{|| \sum_{n=1}^{N} z_n ||}. \tag{6}$$

## 6 METHODOLOGY

We present a method to learn LSBD representations (as defined in Section 3) given some transformation labels, under the assumptions in Section 4. We emphasize that there are two main components that lead to LSBD representations:

1. a suitable topology for the representation space $Z_G$, to be able to model the (linear) action of the underlying symmetry group correctly, and

2. some supervision on transformations to ensure equivariance w.r.t. the predefined group representation.

Our method consists of an unsupervised and a supervised part, such that we can learn from unlabeled data as well as from data with additional supervision on transformations. First we formalize what we mean by transformation labels, then we outline the unsupervised and supervised parts of our method.

**Method dataset: observations with limited transformation labels** For our method we assume access to two disjoint datasets $\mathcal{X}$ and $\mathcal{Y}$ of observations in the data space $X$, where we have additional information on the transformations between elements in $\mathcal{Y}$. More precisely, we assume that $\mathcal{Y}$ consists of $L$ batches of $M$ data points each, where all data points in a batch can be expressed with a *known* group element acting on the first member of the batch. Formally, this dataset can be described as

$$\mathcal{Y} = \{\{x_{lm}: \ x_{lm} = g_{lm} \cdot x_{l1}\}_{m=1}^{M}\}_{l=1}^{L}, \tag{7}$$

where the first group transformation for each batch corresponds to the identity $g_{l1} = e$. The unlabeled dataset $\mathcal{X}$ consists of $N - M \cdot L$ data points with no information about the transformations among the data points.

**Unsupervised learning: VAE with suitable topology** To model data representations for the unlabelled dataset $\mathcal{X}$ in a submanifold $Z_G$ of Euclidean space, we use a Diffusion Variational Autoencoder ($\Delta$VAE) (Perez Rey et al., 2020). This is a Variational Autoencoder (Kingma & Welling, 2014; Rezende et al., 2014) that is capable of encoding data into a closed Riemannian latent space.

$\Delta$VAEs can use any closed Riemannian manifold embedded in a Euclidean space as a latent space (or latent manifold), provided that a certain *projection function* from the Euclidean embedding space into the latent manifold is known and the *scalar curvature* of the manifold is available. The $\Delta$VAE uses a parametric family of posterior approximates obtained from a diffusion process over the latent manifold. To estimate the intractable terms of the negative ELBO, the reparameterization trick is implemented via a random walk.

In the case of $S^1$ as a latent (sub)manifold, we consider the Euclidean space $\mathbb{R}^2$ as the embedding space, and the projection function $\Pi : \mathbb{R}^2 \to S^1$ normalizes points in the embedding space: $\Pi(z) = \frac{z}{|z|}$.[3] The scalar curvature of $S^1$ is 0. It is straightforward to extend these properties to a latent space $S^1 \times S^1$.

**Supervision: enforcing equivariance with transformation labels** Caselles-Dupré et al. (2019) proved that (L)SBD representations cannot be inferred from a training set of unlabeled observations, but that access to the transformations between data points is needed. They therefore use a training set of observation pairs with a given transformation between them.

---

[2]Note that this function is not defined if $|| \sum_{n=1}^{N} z_n || = 0$, but this isn't an issue in practice.
[3]This projection function is not defined for $z = \mathbf{0}$, but this value does not occur in practice.

However, we posit that only a limited amount of supervision is sufficient, if we have access to a larger dataset of unlabeled observations as well. Since obtaining supervision on transformations is typically more expensive than obtaining unsupervised observations, it is desirable to limit the amount of supervision needed.

Therefore, we augment the unsupervised $\Delta$VAE with a supervised method that makes use of the transformation-labeled batches from $\mathcal{Y}$. This method makes use of the same principles from Section 5, but on a batch level instead of a full dataset. By alternating the unsupervised and supervised training phases, we have a method that makes use of both unlabeled and transformation-labeled observations.

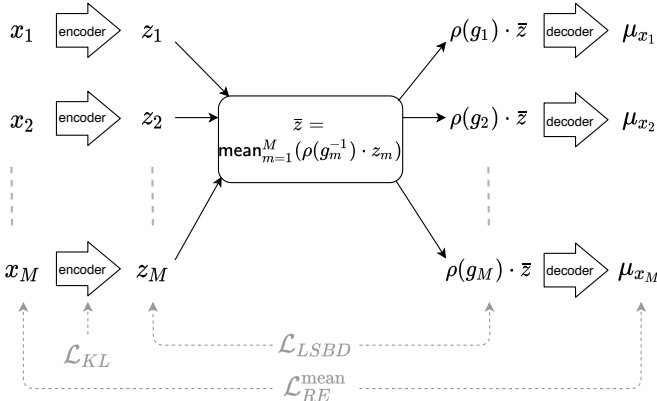

Figure 2: Overview of the computation flow of the supervised training method for a batch $\{x_m\}_{m=1}^M = \{g_m \cdot x_1\}_{m=1}^M$ (where $g_1 = e$).

Figure 2 illustrates the supervised part of our method for a given transformation-labeled batch $\{x_1, x_2, \ldots, x_M\} = \{x_1, g_2 \cdot x_1, \ldots, g_M \cdot x_1\}$ from $\mathcal{Y}$.[4] The supervised part of the method uses the same encoder and decoder networks from the unsupervised part, but an extra processing step is added between encoding and decoding the full batch, which encourages equivariance w.r.t. the given transformation-labels. The procedure of a forward pass is as follows:

1. Compute latent variables $\{z_1, \ldots, z_M\}$, by sampling from the posteriors $q(z|x_m)$ (i.e. a regular forward pass through the encoder and sampling layer of a $\Delta$VAE).
2. Compute $\overline{z} = \text{mean}_{m=1}^M(\rho(g_m^{-1}) \cdot z_m)$.
3. Compute $\rho(g_m) \cdot \overline{z}$ and the parameters of $p(x|\rho(g_m) \cdot \overline{z})$ for (i.e. a regular forward pass of $\rho(g_m) \cdot \overline{z}$ through the $\Delta$VAE decoder).

We train the full network to optimize the following loss function:

$$\mathcal{L}_L = \mathcal{L}_{RE}^{\text{mean}} + \mathcal{L}_{\text{KL}} + \gamma \cdot \mathcal{L}_{LSBD} \tag{8}$$

$$= -\sum_{m=1}^M \log p(x_m|\rho(g_m) \cdot \overline{z}) + \sum_{m=1}^M \text{KL}(q(z|x_m)||p(z)) + \gamma \cdot \sum_{m=1}^M d^2(z_m, \rho(g_m) \cdot \overline{z}). \tag{9}$$

Here $\mathcal{L}_{RE}^{\text{mean}} + \mathcal{L}_{\text{KL}}$ is essentially the regular ELBO used to train a $\Delta$VAE, but instead of a single sample from $q(z|x_m)$ to compute $\mathcal{L}_{RE}$, we used the "corrected" sample $\rho(g_m) \cdot \overline{z}$. This encourages the decoder to follow the required group structure.

## 7  EXPERIMENTS & RESULTS

We designed experiments on datasets with known group decomposition to test the following:

1. Study LSBD representations obtained with our methodology.

---

[4]To avoid notational cluttering we omit the index $l$ describing the batch to which each data point belongs.

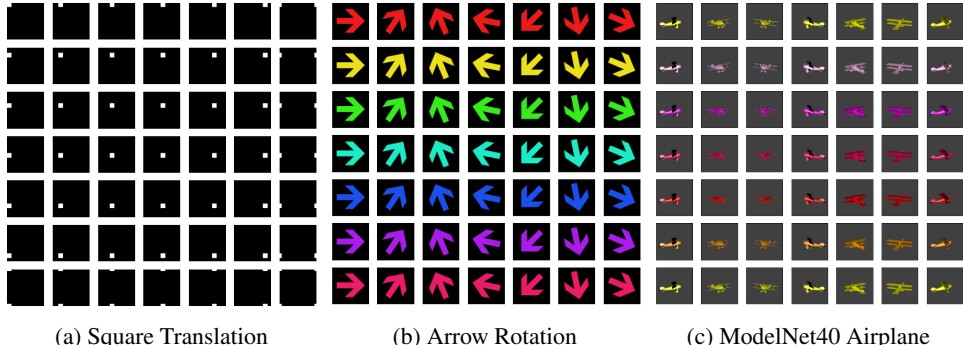

(a) Square Translation      (b) Arrow Rotation      (c) ModelNet40 Airplane

Figure 3: Example images from each of the datasets used. Each image corresponds to an example data point for a combination of two factors, e.g. color and orientation. The factors change horizontally and vertically and the boundaries of each dataset example are periodic.

2. Evaluate the stability and the behavior of the method using our LSBD metric from Section 5 for a range of numbers of labeled pairs $L$ across several training repetitions.

All datasets contain $64 \times 64$ pixel images, with a known group decomposition $G = SO(2) \times SO(2)$ describing the underlying transformations. For each subgroup a fixed number of $64$ transformations is selected. Each image is generated from a single initial data point upon which all possible group actions are applied, resulting in datasets with $N = 4096$ images. The datasets exemplify different group actions of $SO(2)$: periodic translations, in-plane rotations, out-of-plane rotations, and periodic hue-shifts, see Figure 3. For more details, see Appendix E.

For the supervised datasets we use pairs (i.e. batches of size $M = 2$) of transformation-labeled data points. From a dataset of size $N$, we randomly sample $L \leq N/2$ batches of unique data points without replacement. Each pair is labeled with a description of the transformation between the two data points in the pair.

We choose $M = 2$ for our experiments since it is the most limited setting for our supervised method. Higher values of $M$ would provide stronger supervision, so successful results with $M = 2$ imply that good results can also be achieved for higher values of $M$ (but not necessarily vice versa).

In our experiments, we use a fixed weight $\gamma = 100$ and vary the number of batches $L$. We train the $\Delta$VAE for 300 epochs per experiment using the Adam optimizer, each epoch consists of a full unsupervised and supervised phase. For each experiment 10 repetitions were trained and evaluated by computing the LSBD metric across the training dataset. Details of the $\Delta$VAE architecture can be found in Appendix F.

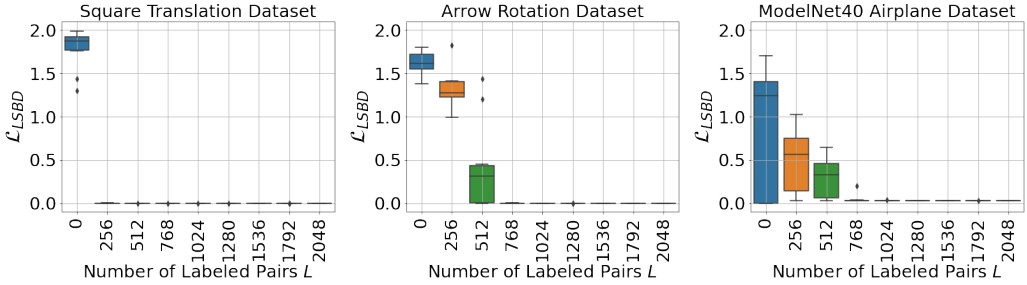

Figure 4: Square Translation, Arrow Rotation and Modelnet40 Airplane datasets' box plots for the LSBD metric evaluated over 10 training repetitions for different numbers of labeled pairs $L$ and $\gamma = 100$.

The results obtained for the three datasets are presented in Figure 4, using the LSBD metric from Section 5 (lower is better). The results show that for all three datasets, only a limited amount of supervision is needed to obtain LSBD representations consistently over multiple runs. In particular,

we observe that by providing transformation labels for disjoint pairs of data points only, our model can learn a more general description of the underlying transformations that connect the entire dataset.

## 8 CONCLUSION & FUTURE WORK

In this work, we specified some well-defined assumptions that allow us to define an easy-to-compute metric to quantify Linearly Symmetry-Based Disentanglement (LSBD), as defined by Higgins et al. (2018). Moreover, we presented a method to obtain such LSBD representations, making use of unlabeled observations as well as a limited amount of supervision on transformations.

Using our LSBD metric, we showed experimentally that our method can indeed learn LSBD representations, using only a limited amount of supervision on transformations, by making use of unlabeled observations as well.

Our LSBD metric and method require a number of assumptions, as explained in Section 4. This limits the applicability of the metric and method, but also provides a clear direction towards a more general approach; by studying how we can relax these well-formalized assumptions.

Moreover, our metric is in fact an upper bound to a more general metric (see Appendix D), which is however less straightforward to compute. Characterizing how to compute this metric in various situations will allow for better comparison with other methods aiming to obtain LSBD representations.

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

APPENDIX

## A   PRELIMINARIES: GROUP THEORY

In this appendix, we summarize some concepts from group theory that are important to understand the main text of the paper. Group theory provides a useful language to formalize the notion of symmetry transformations and their effects. For a more elaborate discussion we refer the reader to the book from Hall (2015) on group theory.

**Group**   A group is a non-empty set $G$ together with a binary operation $\circ : G \times G \to G$ that satisfies three properties:

1. *Associativity*: For all $f, g, h \in G$, it holds that $f \circ (g \circ h) = (f \circ g) \circ h$.
2. *Identity*: There exists a unique element $e \in G$ such that for all $g \in G$ it holds that $e \circ g = g \circ e = g$.
3. *Inverse*: For all $g \in G$ there exists an element $g^{-1} \in G$ such that $g^{-1} \circ g = g \circ g^{-1} = e$.

**Direct product**   Let $G$ and $G'$ be two groups. The *direct product*, denoted by $G \times G'$, is the group with elements $(g, g') \in G \times G'$ with $g \in G$ and $g' \in G'$, and the binary operation $\circ : G \times G' \to G \times G'$ such that $(g, g') \circ (h, h') = (g \circ h, g' \circ h')$.

**Lie group**   A Lie group is a group where $G$ is a smooth manifold, this means it can be described in a local scale with a set of continuous parameters and that one can interpolate continuously between elements of $G$.

**Group action**   Let $A$ be a set and $G$ a group. The *group action* of $G$ on $A$ is a function $G_A : G \times A \to A$ that has the properties [5]

1. $G_A(e, x) = x$ for all $a \in A$
2. $G_A(g, (G_A(g', a)) = G_A(g \circ g', a)$ for all $g, g' \in G$ and $a \in A$

**Regular action**   The action of $G$ on $A$ is regular if for every pair of elements $a, a' \in A$ there exists a unique $g \in G$ such that $g \cdot a = a'$.

**Group representation**   A *group representation* of $G$ in the vector space $V$ is a function $\rho : G \to GL(V)$ (where $GL(V)$ is the general linear group on $V$) such that for all $g, g' \in G$ $\rho(g \circ g') = \rho(g) \circ \rho(g')$ and $\rho(e) = \mathbb{I}_V$, where $\mathbb{I}_V$ is the identity matrix.

**Direct sum of representations**   The direct sum of two representations $\rho_1 : G \to GL(V)$ in $V$ and $\rho_2 : G \to GL(V')$ in $V'$ is a group representation $\rho_1 \oplus \rho_2 : G \to GL(V \oplus V')$ over the direct sum $V \oplus V'$, defined for $v \in V$ and $v' \in V'$ as:

$$(\rho_1 \oplus \rho_2)(g) \cdot (v, v') = (\rho_1(g) \cdot v, \rho_2(g) \cdot v') \tag{10}$$

## B   SYMMETRY-BASED DISENTANGLEMENT DEFINITION

Given the assumptions stated in Section 3, the definition for Symmetry-Based Disentangled (SBD) representations introduced by (Higgins et al., 2018) is as follows:

**Symmetry-Based Disentangled (SBD) Representations**   Suppose that the group $G$ is decomposed as the direct product of $K$ groups, $G = G_1 \times \ldots \times G_K$ which models the transformations of the world states via an action $\cdot : G \times W \to W$ on $W$. A representation $f$ is a SBD representation with respect to $G = G_1 \times \ldots \times G_K$ if

---

[5]To avoid notational clutter, we write $G_A(g, a) = g \cdot a$ where the set $A$ on which $g \in G$ acts can be inferred from the context.

- there is an action $\cdot : G \times Z \to Z$,

- the map $f : W \to Z$ is *equivariant* with respect to the actions of $G$ on $W$ and $Z$, i.e. $\forall w \in W$ and $g \in G$ it holds that $f(g \cdot w) = g \cdot f(w)$, and

- there is a decomposition $Z = Z_1 \times \ldots \times Z_K$ such that each $Z_i$ is invariant to the action of all but one subgroup. This means that for $z_i \in Z_i$ $g \cdot z_i = z_i$ for all $g \in G_j$ if $j \neq i$ and it is only affected by $G_i$.

## C  Symmetry-Based Disentanglement in the data space

If $b$ is injective, then we can define an action of $G$ on the possible observations $b(W) \subseteq X$. The injectivity of $b$ assures unique observations for different world states (in cases where $b$ is not injective, it is possible to address this through active sensing (Soatto, 2011), so assuming injectivity is reasonable in practice). If $b$ is injective then restricted to its image it is bijective and we can define the action $\cdot : G \times b(W) \to b(W)$ of $G$ on $b(W)$ as $g \cdot x = b(g \cdot b^{-1}(x))$.

If the action of $G$ on $W$ is *regular* then $b$ is equivariant with respect to the actions of $G$ on $W$ and $b(W)$. In such case, if $h$ is LSBD with respect to the decomposition of $G$ for the available data then $f$ is also LSBD. This means that in such a setting we only need to focus on the disentanglement of the inference map $h$, as this will assure the disentanglement of the model's representation $f$ of the world.

## D  General metric

In this appendix, we show how the disentanglement metric $\mathcal{L}_{LSBD}$ in the main text (with a specific choice for the metric $d$) is an upper bound for a more general LSBD metric $\mathcal{M}_{LSBD}$. The more general metric can be defined for any encoding map from data space $X$ to latent space $Z$ and a data probability measure $\mu$ on $X$ if $\mu$ satisfies the following condition. We assume the data (probability) measure $\mu$ equals the pushforward $G_X(\cdot, x_1)_{\#}\nu$ of a probability measure $\nu$ on $G$ by the function $G_X(\cdot, x_1)$, i.e.

$$\mu(A) = (G_X(\cdot, x_1)_{\#}\nu)(A) = \nu\left(\{g \in G \mid G_X(g, x_1) \in A\}\right)$$

for Borel subsets $A \subset X$. Here $G_X$ denotes the action of $G$ on $X$ and $x_1$ is a base point. The situation of a discrete dataset $\{x_n\}_{n=1}^N = \{g_i \cdot x_1\}_{n=1}^N$ corresponds to the case in which $\nu$ and $\mu$ are following empirical measures on the group $G$ and data space $X$ respectively

$$\nu := \frac{1}{N} \sum_{i=1}^N \delta_{g_i} \qquad \mu := \frac{1}{N} \sum_{i=1}^N \delta_{x_i}.$$

We define the more general LSBD metric $\mathcal{M}_{LSBD}$ as follows

$$\mathcal{M}_{LSBD} :=$$
$$\inf_{\rho \in \mathcal{P}(G,Z)} \int_G \left\| \rho(g)^{-1} \cdot h(g \cdot x_1) - \int_G \rho(g')^{-1} \cdot h(g' \cdot x_1) d\nu(g') \right\|_{\rho,h,\mu}^2 d\nu(g) \tag{11}$$

where the norm $\| \cdot \|_{\rho,h,\mu}$ is a Hilbert-space norm depending on the representation $\rho$, the encoding map $h : X \to Z$, and the data measure $\mu$. Moreover, $\mathcal{P}(G, Z)$ denotes the set of linear representations of $G$ in $Z$. From now on, we will denote the norm as $\| \cdot \|_*$.

To describe the norm $\| \cdot \|_*$ we start with an arbitrary inner product $(\cdot, \cdot)$ on the linear latent space $Z$. Assume that $\rho$ splits in irreducible representations $\rho_i : G \to Z_i$ where $Z = Z_1 \oplus \cdots \oplus Z_k$ for some $k \in \mathbb{N}$. We will define a new inner product $\langle \cdot, \cdot \rangle_*$ on $Z$ as follows. First of all we declare $Z_i$ and $Z_j$ to be orthogonal with respect to $\langle \cdot, \cdot \rangle_*$ if $i \neq j$. We denote by $\pi_i$ the orthogonal projection on $Z_i$.

For $v, w \in Z_i$, we set

$$\langle v, w \rangle := \lambda_i^{-1} \int_{g \in G} (\rho(g) \cdot v, \rho(g) \cdot w) d\mathfrak{m}(g) \tag{12}$$

where $\mathfrak{m}$ is the (bi-invariant) Haar measure normalized such that $\mathfrak{m}(G) = 1$ and set

$$\lambda_i := \int_X \int_G \|\pi_i(h(x))\|^2 d\mathfrak{m}(g) d\mu(x) \tag{13}$$

if the integral on the right-hand side is strictly positive and otherwise we set $\lambda_i := 1$. This construction completely specifies the new inner product, and it has the following properties:

- the subspaces $Z_i$ are mutually orthogonal
- $\rho_i(g)$ is orthogonal on $Z_i$ for every $g \in G$, in other words $\rho_i$ maps to the orthogonal group on $Z_i$. Moreover, $\rho$ maps to the orthogonal group on $Z$. This follows directly from the bi-invariance of the Haar measure and the definition of $(\cdot, \cdot)_*$.
- If $\pi_i$ is the orthogonal projection to $Z_i$, then

$$\int_X \|\pi_i(h(x))\|_*^2 d\mu(x) = 1 \tag{14}$$

if the integral on the left is strictly positive.

We now explain why $\mathcal{L}_{LSBD}$ is an upper bound for $\mathcal{M}_{LSBD}$, in case $Z$ in the main text is endowed with the norm $\|\cdot\|_*$. Note also that in the example with $SO(2)$ rotations in the main text, the norm $\|\cdot\|_*$ is just the ordinary Euclidean norm.

First of all, because of the infimum in the definition of $\mathcal{M}_{LSBD}$, for any representation $\rho$ considered in the main text we find

$$\mathcal{M}_{LSBD} \leq \int_G \left\| \rho(g)^{-1} \cdot h(g \cdot x_1) - \int_G \rho(g')^{-1} \cdot h(g' \cdot x_1) d\nu(g') \right\|_*^2 d\nu(g) \tag{15}$$

Finally, it is a property of the mean that for every $c \in Z$ the right-hand side is smaller than

$$\int_G \left\| \rho(g)^{-1} \cdot h(g \cdot x_1) - c \right\|_*^2 d\nu(g). \tag{16}$$

Written out for a discrete dataset, letting $z_n := h(x_n)$, and taking the mean from Equation (6)

$$c := \operatorname*{mean}_{n=1}^N (z_n) =: \overline{z} \tag{17}$$

we find that

$$\mathcal{M}_{LSBD} \leq \sum_{n=1}^N \left\| \rho(g_n)^{-1} \cdot z_n - \overline{z} \right\|_*^2 = \mathcal{L}_{LSBD}. \tag{18}$$

We will now give an alternative expression for the disentanglement metric $\mathcal{M}_{LSBD}$, since it will more visibly relate to the definition of equivariance. Let $\rho \in \mathcal{P}(G, Z)$ be a linear representation of $G$ in $Z$. By expanding the inner product (or by using usual computation rules for expectations and variances), we first find that

$$\int_G \left\| \rho(g)^{-1} \cdot h(g \cdot x_1) - \int_G \rho(g')^{-1} \cdot h(g' \cdot x_1) d\nu(g') \right\|_*^2 d\nu(g)$$
$$= \int_G \left\| \rho(g)^{-1} \cdot h(g \cdot x_1) \right\|_*^2 d\nu(g) - \left\| \int_G \rho(g)^{-1} \cdot h(g \cdot x_1) d\nu(g) \right\|_*^2 \tag{19}$$
$$= \frac{1}{2} \int_G \int_G \left\| \rho(g)^{-1} \cdot h(g \cdot x_1) - \rho(g')^{-1} \cdot h(g' \cdot x_1) \right\|_*^2 d\nu(g) d\nu(g').$$

We now use that $\rho$ maps to the orthogonal group for $(\cdot, \cdot)_*$, so that we can write the same expression as

$$\frac{1}{2} \int_G \int_G \left\| \rho(g \circ g'^{-1})^{-1} \cdot h(((g \circ g'^{-1}) \cdot g') \cdot x_1) - h(g' \cdot x_1) \right\|_*^2 d\nu(g) d\nu(g') \tag{20}$$

This brings us to the alternative characterization of $\mathcal{M}_{LSBD}$ as

$$\mathcal{M}_{LSBD} = \inf_{\rho \in \mathcal{P}(G, Z)} \frac{1}{2} \int_G \int_G \| \rho(g \circ g'^{-1})^{-1} h(((g \circ g'^{-1}) \cdot g') \cdot x_1) - h(g' \cdot x_1) \|_*^2 d\nu(g) d\nu(g') \tag{21}$$

In particular, if for every data point $x$ there is a unique group element $g_x$ such that $x = g_x \cdot x_1$, the disentanglement metric $\mathcal{M}_{LSBD}$ can also be written as

$$\inf_{\rho \in \mathcal{P}(G,Z)} \frac{1}{2} \int_G \int_X \|\rho(g \circ g_x^{-1})^{-1} h((g \circ g_x^{-1}) \cdot x) - h(x)\|_*^2 d\nu(g) d\mu(x) \tag{22}$$

in which the equivariance condition appears prominently. The condition becomes even more apparent if $\nu$ is in fact the Haar measure itself, in which case the metric equals

$$\inf_{\rho \in \mathcal{P}(G,Z)} \frac{1}{2} \int_G \int_X \|\rho(g)^{-1} \circ h(g \cdot x) - h(x)\|_*^2 d\mathfrak{m}(g) d\mu(x) \tag{23}$$

Although this expression would also make for a natural choice of metric in general (when $\nu$ is not necessarily the Haar measure), this choice is not directly practical as one does not have access to data points $g \cdot x$ for arbitrary $g$.

## E  DATASETS: ADDITIONAL INFORMATION

**Square Translation**    This dataset consists of a set of images of a black background with a square of $8 \times 8$ white pixels. The dataset is generated applying vertical and horizontal translations of the white square considering periodic boundaries.

**Arrow Rotation**    This dataset consists of a set of images depicting a colored arrow at a given orientation. The dataset is generated by applying cyclic shifts of its color and in-plane rotations. The cyclic color shifts were obtained by preselecting a fixed set of $64$ colors from a circular hue axis. The in-plane rotations were obtained by rotating the arrow along an axis perpendicular to the picture plane over $64$ predefined positions.

**ModelNet40 Airplane**    The ModelNet40 Airplane dataset consists of a dataset of renders obtained using Blender v2.7 Community (2020) from a 3D model of an airplane within the ModelNet40 dataset Wu et al. (2014); Sedaghat & Brox (2015). We created each image by varying two properties: the airplane's color and its orientation with respect to the camera. The orientation was changed via rotation with respect to a vertical axis (out-of-plane rotation). The colors of the model were selected from a predefined cyclic set of colors similar to the arrow rotation dataset.

## F  $\Delta$VAE ARCHITECTURE

Table 1: Encoder architecture

| Input shape | Layer | Output shape |
|---|---|---|
| (64, 64, d) | Conv2D | (64, 64, 64) |
| (64, 64, 64) | MaxPool2D | (32, 32, 64) |
| (32, 32, 64) | Conv2D | (32, 32, 64) |
| (32, 32, 64) | MaxPool2D | (16, 16, 64) |
| (16, 16, 64) | Conv2D | (16, 16, 64) |
| (16, 16, 64) | MaxPool2D | (8, 8, 64) |
| (8, 8, 64) | Flatten | 8*8*64 |
| 8*8*64 | Dense | 64 |
| 64 | Dense | 4+1 |

The architecture of the $\Delta$VAE encoder and decoder are shown in Tables 1 and 2, respectively. For grayscale images (Square Translation), $d = 1$, for RBG images, $d = 3$. All convolutional layers use a kernel size of $3 \times 3$ with 64 filters, stride 1, and are followed by a ReLU activation (with the exception of the final decoder layer, which uses a sigmoid activation to produce pixel values between 0 and 1). MaxPooling layers use have pool size of $2 \times 2$. The number of units in the dense (or fully connected) layers is given by their output shape. Dense layers are followed by a ReLU activation, except for the last layer of the encoder. UpSampling layers enlarge the first two dimensions by a factor of 2, using nearest neighbor interpolation.

Table 2: Decoder architecture

| Input shape | Layer | Output shape |
|---|---|---|
| 4 | Dense | 64 |
| 64 | Dense | 8*8*64 |
| 8*8*64 | Reshape | (8, 8, 64) |
| (8, 8, 64) | UpSampling2D | (16, 16, 64) |
| (16, 16, 64) | Conv2D | (16, 16, 64) |
| (16, 16, 64) | UpSampling2D | (32, 32, 64) |
| (32, 32, 64) | Conv2D | (32, 32, 64) |
| (32, 32, 64) | UpSampling2D | (64, 64, 64) |
| (64, 64, 64) | Conv2D | (64, 64, d) |

The first 4 dimensions of the encoder output are projected onto the latent manifold $S^1 \times S^1$ to represent the mean parameter $\mu$ of the posterior, the final dimension represents the parameter $\log t$, for which we limit the values between -10.0 and -5.0 for numerical stability. See Perez Rey et al. (2020) for more details about these parameters and the sampling process.

