# OpenReview forum: "Quantifying and Learning Disentangled Representations with Limited Supervision"
_ICLR.cc/2021/Conference — Reject_

### Official Review · AnonReviewer2 · 2020-10-26
**Needs comparisons with other disentangling metrics/methods, and explanations for how the method can scale to realistic datasets**

**Rating:** 5
**Confidence:** 3

**Review:**

Summary: This paper aims to operationalize the symmetry-based disentanglement idea proposed in Higgins et al., 2018, and proposes a novel loss function that can be used to both evaluate and learn disentangled representations (including learning from datasets where only a subset of samples are fully labeled). The paper includes some experimental validation of the proposed ideas on synthetic datasets.

Overall, the paper is clearly written and makes a positive contribution in pushing for disentangling metrics that are motivated by better formalism than what is commonly used. The paper would be stronger if it included rigorous comparisons to state of the art disentangling metrics and methods, and a discussion of how the requirement of full supervision of all factors of variation for at least a portion of the dataset can be relaxed.

Weaknesses:
* The supervised subset mentioned in section 6 must have full labels for all factors/groups. If even one factor is missing, then this method cannot be applied, since we do not know the full set of transformations that relate different data points to each other. It is not clear to me how this assumption can be relaxed with the proposed metric/objective: if we lack labels for only a single factor, then it is no longer possible to explain the dataset in terms of actions on a single input sample.
* The above also means that experiments can only be done on synthetic datasets, where all factors are known.
* This paper would be stronger if it (a) showed a comparison of the LSBD metric to commonly used metrics for disentangling, such as those compared in Locatello et al., 2018. and (b) compared the proposed model to other disentangling models, both supervised and unsupervised. An implicit comparison to diffusion VAE can be found in the results in Figure 4, when the number of labeled pairs is zero, but a more thorough comparison is required to understand the value of the model and metric.

Strengths:
* Novel method for evaluating and learning disentangled representations.
* Can leverage unlabeled data.

Clarity
* In the experiments, the g’s are given by the data. But how are the \rho functions implemented? Are they specified in advance?
Section 4, bullet point 3: “each observation is a transformed version of another observation, and that transformation is unique”. This seems like a fairly strong statement that is often violated: for example, rotating and flipping a square can easily map to the same X. It would be useful to discuss:
- how does the metric/model behave when these assumptions are violated in a domain?
- how does the loss react to subspaces that are scaled differently, or subspaces of differing dimensionality? In the experiments I believe all the subspaces are the  same size, but if one was in R^2 and another in R^100, the squared-distances would be dominated by the latter and the loss may not work as well.

---

> ### Author Response · Authors · 2020-11-21
> **Author response (1/2)**
>
> Dear reviewer, we would like to thank you for the time and effort spent in analyzing this paper and for the specific suggestions made to improve this work. We will address the concerns presented in the review in the same order as presented.
>
> (1) *The supervised subset mentioned in section 6 must have full labels for all factors/groups. If even one factor is missing, then this method cannot be applied, since we do not know the full set of transformations that relate different data points to each other. It is not clear to me how this assumption can be relaxed with the proposed metric/objective: if we lack labels for only a single factor, then it is no longer possible to explain the dataset in terms of actions on a single input sample.The above also means that experiments can only be done on synthetic datasets, where all factors are known.*
>
> It is true that the supervised training of Section 6 requires full labels for all groups, if one group of interest is not available as a label for a datapoint, that datapoint can't be used for supervised training. This means that the task of labeling can be intensive. However, our experiments on the amount of supervision show that disentanglement can already be achieved with a limited amount of labels.
>
> We do agree that it is an interesting problem if only partial labels are available (where only some subgroup values of a transformation are known). However, this is beyond the scope of our paper, and we believe that our work provides a good basis for addressing such a problem setting. Moreover, we think there are many practical settings in which full transformation labels can be obtained, for a limited amount of data (sufficient to achieve disentanglement).
>
>
> (2) *This paper would be stronger if it (a) showed a comparison of the LSBD metric to commonly used metrics for disentangling, such as those compared in Locatello et al., 2018. and (b) compared the proposed model to other disentangling models, both supervised and unsupervised. An implicit comparison to diffusion VAE can be found in the results in Figure 4, when the number of labeled pairs is zero, but a more thorough comparison is required to understand the value of the model and metric.*
>
> We emphasize that we focus on symmetry-based disentanglement (Higgins et al., 2018), whereas previous work such as in Locatello et al., 2018 uses a factor-based definition of disentanglement. These definitions have different implications, in particular LSBD requires certain properties from disentangled representations that will simply not arise in previous disentanglement methods. Therefore we believe a comparison would be unfair (favoring our method on the LSBD metric). Vice versa it is similar; previous disentanglement metrics typically want a single latent dimension to encode a single generative factor, whereas we show that LSBD requires at least 2-dimensional subspaces for SO(2) group structures. Existing disentanglement metrics cannot deal with this, or would need to be adjusted.
>
> We do believe further study into the differences between factor-based and symmetry-based disentanglement is a very interesting and useful topic, but it is outside of the scope of this paper. At this point we believe a decent comparison with other methods cannot be made, and would therefore only be misleading. Instead our experimental results serve as a sanity check for our LSBD metric, and as an investigation into the effect of supervision.
>
>
>
> (3) *In the experiments, the g’s are given by the data. But how are the \rho functions implemented? Are they specified in advance?*
>
> In our method we indeed consider that the \rho functions are specified in advance, as per our assumptions in Section 4 (assumption #4). We agree that this is a limitation of our current method/metric, but as with all of the assumptions in Section 4 we believe it provides a good starting point towards a more general method/metric, by investigating how to relax these (well-formalized) assumptions
>
>
> ... to be continued (1/2)

---

> > ### Author Response · Authors · 2020-11-21
> > **Author response (2/2)**
> >
> > (4)*Section 4, bullet point 3: “each observation is a transformed version of another observation, and that transformation is unique”. This seems like a fairly strong statement that is often violated: for example, rotating and flipping a square can easily map to the same X. It would be useful to discuss:*
> >
> > *A)how does the metric/model behave when these assumptions are violated in a domain?*
> >
> > We recognize the importance of analyzing these cases that do not fulfill the assumptions considered in this paper, however we consider these situations out of scope for the paper since our goal is to establish a starting point for the measurement of LSBD in simple situations.
> >
> >
> > *B)how does the loss react to subspaces that are scaled differently, or subspaces of differing dimensionality? In the experiments I believe all the subspaces are the same size, but if one was in R^2 and another in R^100, the squared-distances would be dominated by the latter and the loss may not work as well.*
> >
> > In this situation the loss would indeed give different weight to each of the subspaces (when using Euclidean distance). A certain weight factor could be added to the loss to alleviate this problem. In our more general formulation of the LSBD metric in the appendix, we address this with normalization in the definition of the inner product that is used to define the distance metric.

---

### Official Review · AnonReviewer4 · 2020-10-27

**Rating:** 4
**Confidence:** 3

**Review:**

=============================================================================================================

Summary:
The paper starts from the Linear Symmetry-Based Disentanglement (LSBD) in [1]. The existing approaches to evaluate disentanglement require the supervision of the dataset and it is impossible for the unlabeled data. In this regard, the authors propose the new quantifying metric for disentanglement under the 'limited supervision' setting. Also, under this metric, authors provide a VAE-based framework that exploits limited supervision for the proposed metric.

==============================================================================================================

Reason for score:

Overall, I vote for clear rejection. I was hooked by the title and abstract. However, I could not find any novelty and insights to compare several measures for the disentanglement score. Furthermore, experiments are not enough to support the proposed method (see cons). I suggest the resubmission with more experiment results and justifications of their proposed methods.

=============================================================================================================

Strong points (pros) :

(1) Mathematical definitions to interpret disentanglement are clear (but complex). The quality of writing is not bad.

(2) I like the trial to quantify disentanglement under a limited-supervised setting since it could be applied to real-world settings.

==============================================================================================================

cons :

(1) The comparison with the other disentanglement metrics under supervised data is crucial. The proposed metric which can be applied under a weakly supervised setting should have a certain amount of consensus with other metrics. The authors need to show through experiments and provide justification by pointing out similarities and differences with the other metrics. This part is very important to persuade reviewers and readers.

(2) I was caught off guard at the experiments since there doesn't exist any baselines in Figure 4. The authors need to add baselines to compare with $\Delta VAE$.

(3) The idea underlying paper is quite overlapped with [2]. As a baseline, [2] can be used to analyze the performance of $\Delta VAE$. Also, the authors need to emphasize the novelty of their method by comparing it with [2].

=====================================================================================================

Minor :

The word "limited supervision" is quite confusing. Instead, I suggest "weakly-supervised".

======================================================================================================

After rebuttal :

Thank you for the responses. I'm not sure why the authors didn't perform the experiments on the correlation between the previous factor-based disentanglement scores and the proposed disentanglement score in the limited supervised setting. For example, if there are 5 factors, I propose to evaluate the proposed disentanglement score for every possible pair of the factors (10 pairs) and average the scores. I believe this paper handles the valuable topic but it is not enough to be accepted since the experiments, which are crucial I believe, are omitted (comparison with the other disentanglement score, baselines to $Delta$VAE, comparison with [2]). Also, I concerned that other readers might be confused with the ("factor-based" disentanglement and "symmetric-based" disentanglement ) and (limited-supervision and weakly-supervision) (a new section should be added to handle these topics if this paper should be accepted). Furthermore, discussion with the related works is not enough.


I lower my confidence rate to 3 (5->3) and vote for weak reject (3->4). But, I hope this paper would be accepted after revisions in the future.

============================================================================================================

References

[1] beta-VAE: Learning Basic Visual Concepts with a Constrained Variational Framework, ICLR 2017.

[2] Weakly-Supervised Disentanglement Without Compromises, ICML2020.

---

> ### Author Response · Authors · 2020-11-21
> **Author response (1/2)**
>
> Dear reviewer, we would like to thank you for the time and effort spent in analyzing this paper and for the specific suggestions made to improve this work. We will address the concerns presented in the review in the same order as presented.
>
>
> (1) *The comparison with the other disentanglement metrics under supervised data is crucial. The proposed metric which can be applied under a weakly supervised setting should have a certain amount of consensus with other metrics. The authors need to show thorough experiments and provide justification by pointing out similarities and differences with the other metrics. This part is very important to persuade reviewers and readers.*
>
> Firstly, to avoid any confusion, we would like to stress that we are not basing ourselves on the beta-VAE paper [1] (Higgins et al., 2017), which deals with the more common "factor-based" disentanglement, but on "Towards a Definition of Disentangled Representations" (Higgins et al., 2018), which proposed "symmetry-based" disentanglement. The symmetry-based approach to disentanglement is significantly different from most factor-based works, and as such the possibility of comparison with such methods is limited. In particular, most factor-based methods and metrics require disentanglement between single latent dimensions, whereas our analysis shows that Linear Symmetry-Based Disentanglement may require higher-dimensional subspaces (e.g. in the case of SO(2) structures at least 2-dimensional) to be disentangled. Existing disentanglement metrics cannot deal with this, or would need to be adjusted.
>
> Our claim for novelty is not in incremental improvements on some established metrics, but in advancing a different perspective on disentanglement, LSBD; by showing how to quantify such disentanglement and by showing a method to obtain LSBD representations. We believe that the justification of LSBD follows from its formal definition, and that the similarities and differences between factor-based and symmetry-based disentanglement are clear from their definitions. Our aim is then to provide a quantification of LSBD that is justified through mathematical formalism, rather than through comparisons with older disentanglement metrics that lack such formalism.
>
> We do believe further study into the differences between factor-based and symmetry-based disentanglement is a very interesting and useful topic, but it is outside of the scope of this paper. Furthermore, a lot of work can be done on generalizing the quantification of LSBD, and on generalizing the methodology to obtain LSBD representations. We do not claim to provide a full solution to this, but we take an important step in this direction by clearly formalizing the assumptions we take in this work. Future work in this direction then only needs to focus on how to relax these assumptions.
>
>
> (2) *I was caught off guard at the experiments since there doesn't exist any baselines in Figure 4. The authors need to add baselines to compare with ΔVAE.*
>
> As mentioned above, our contribution is not focused on improvement over some existing baseline, but on building on top of the formal framework of LSBD by providing a quantification, and a method to obtain LSBD representations. Figure 4 focuses on showing the influence of weak supervision on disentanglement results. We reason why LSBD cannot be obtained without the proper latent manifold, so any baseline without the right manifold structure would make an unfair comparison (favoring our method) and is therefore left out. Note that the unsupervised ΔVAE is not part of our contribution, we only propose the supervised extension. Thus, ΔVAE (without supervision) is itself indeed a baseline.
>
>
> (3) *The idea underlying paper is quite overlapped with [2]. As a baseline, [2] can be used to analyze the performance of ΔVAE. Also, the authors need to emphasize the novelty of their method by comparing it with [2].*
>
> Thank you for this reference, we were unaware of this paper. However, we emphasize again that we focus on symmetry-based disentanglement (Higgins et al., 2018), whereas [2] deals with factor-based disentanglement. Our novelty is in advancing the symmetry-based framework, for which we indeed show the influence of weak supervision. This does not directly compare to weak supervision for factor-based disentanglement. Although we believe that further studies into comparing these two frameworks are an interesting topic, it is beyond the scope of our paper.
>
>
> ... to be continued (1/2)

---

> > ### Author Response · Authors · 2020-11-21
> > **Author response (2/2)**
> >
> > *The word "limited supervision" is quite confusing. Instead, I suggest "weakly-supervised".*
> >
> > Thank you for this advice, we see that it is indeed confusing. We used the term limited supervision to indicate that small amounts of supervised information suffice to obtain LSBD. Moreover, we wanted to emphasise that our supervision is in fact on transformations, not on direct values as is most common in ML. We seriously considered changing to the term "weak supervision", but after some research we conclude that there is no agreement on the meaning of this term either, and that in particular it is often used an an umbrella term that includes limited supervision, besides other concepts that are not applicable to our work (such as imprecise/inexact labels). Therefore, although we agree there can be some confusion, we hope that "limited supervision" best captures our situation, even though there is no clear agreement on the terminology within the community.

---

### Official Review · AnonReviewer1 · 2020-10-28
**An interesting metric of disentanglement, but claims and technical aspects need to be clarified**

**Rating:** 5
**Confidence:** 3

**Review:**

The paper contributes an interesting new measure for disentanglement which consists in: (1) applying to each latent representation a (supervised) linear transformation mapping it to the origin representation (2) measuring the average distance of these canonicalized representation to the origin to obtain a mean discrepancy measure of disentanglement. By adding this measure as an additional loss term in a topological VAE, the authors show that the model can learn disentangled representations with a limited amount of supervision on some simple image datasets.

Strong points:
- the measure of disentanglement proposed is promising and novel.
- the mathematical derivations and the simulation results are convincing.

Weak points:
- some claims in the paper are misleading because they imply that the measure proposed is unsupervised, although it is totally supervised (see below for details).
- some technical aspects of the paper lack clarity (see below for details).
- the measure of disentanglement proposed requires a lot of prior knowledge on the data and on the structure of the representation: for all pairs of data points on which the measure is calculated, we need to know what group element this transformation corresponds to, and we need to know to which linear transformation this group element corresponds to in the latent representation. This is impractical in many datasets where the transformations are unknown, and in many models where the latent equivariant operators are learned and not pre-specified.

I recommend to reject this paper due to the clarity concerns and concerns about the validity of the claims, unless they can be addressed satisfactorily during the rebuttal period.

Main concerns to be addressed:
1) It is impossible to understand from the abstract, intro and related work section whether the proposed measure of disentanglement requires supervision about the transformation between pairs of data points or not. In fact, the measure is 100% supervised. I agree that the model proposed using this measure as an additional loss term is only supervised on part of the data, but the measure itself is supervised. Here are examples of misleading claims:
-Abstract: "Although several works focus on learning LSBD representations, none of them provide a metric to quantify disentanglement. Moreover, such methods require supervision on the underlying transformations for the entire dataset, and cannot deal with unlabeled data."
-Related Work: "Moreover, their methods require supervision on the transformation relationships among datapoints for the entire training dataset."
2) Some technical aspects of the paper are unclear:
- LSBD is clearly defined, SBD is mentioned multiple times but never defined.
- the ∆VAE model is never described, nor its architecture. The reader is required to read another paper to understand how this model works.
- "An easy-to-compute metric to quantify LSBD given certain assumptions (see Section 4), which acts as an upper bound to a more general metric (derived in Appendix C)." This more general metric is not described in the main text, and it is unclear what the underlying motivation is for this alternative metric (and reading the technical description in the appendix did not help me understand the motivation).
- Enigmatic discussion points: "Our LSBD metric and method require a number of assumptions, as explained in Section 4. This limits the applicability of the metric and method, but also provides a clear direction to what needs to be done to obtain and quantify LSBD representations if these assumptions are relaxed.", "Moreover, our metric is in fact an upper bound to a more general metric (see Appendix C), which is however less straightforward to compute." What are the clear directions on what needs to be done? How to think of this more general metric?

Additional feedback:
- The LSBD assumptions at restated three times in the main text (p2,3,4), which is unnecessarily redundant.

---

> ### Author Response · Authors · 2020-11-21
> **Author response (1/2)**
>
> Dear reviewer, we would like to thank you for the time and effort spent in analyzing this paper and for the specific suggestions made to improve this work. We will address the concerns presented in the review, which we enumerate here:
>
>
> (1) *The measure of disentanglement proposed requires a lot of prior knowledge on the data and on the structure of the representation: for all pairs of data points on which the measure is calculated, we need to know what group element this transformation corresponds to, and we need to know to which linear transformation this group element corresponds to in the latent representation. This is impractical in many datasets where the transformations are unknown, and in many models where the latent equivariant operators are learned and not pre-specified.*
>
> We recognize that our requirements for the prior knowledge can be restrictive, however we believe that they are a starting point for a discussion on a general metric for Linear-Symmetry Based Disentanglement (LSBD), which, to the best of our knowledge, is still missing. In comparison to other work on disentanglement, we would like to point out that other disentanglement techniques and metrics (not necessarily for LSBD) also require strong assumptions on the prior knowledge on data used for evaluation as it has been pointed out also in previous work by Locatello *et al.*2019 [1]. In particular, to the best of our knowledge, all existing disentanglement metrics also require full supervised knowledge on the generative factors. We recognize the importance of reducing these requirements in general, but we consider this a challenge that should be addressed in future work (but which we start adressing already in Appendix C).
>
>
> (2) *In fact, the measure is 100% supervised. I agree that the model proposed using this measure as an additional loss term is only supervised on part of the data, but the measure itself is supervised. Here are examples of misleading claims:
> -Abstract: "Although several works focus on learning LSBD representations, none of them provide a metric to quantify disentanglement. Moreover, such methods require supervision on the underlying transformations for the entire dataset, and cannot deal with unlabeled data." -Related Work: "Moreover, their methods require supervision on the transformation relationships among datapoints for the entire training dataset."*
>
> We appreciate the feedback on clarity and we will update the text to makes this clearer. We are indeed considering a measure that is completely supervised and a method of disentanglement that is semi-supervised. Note the distinction between the *method* (a training method to obtain disentangled representations in practice, semi-supervised) and the *metric* (a way to quantify LSBD, for evaluation in academic works, fully supervised). We remark that this is a common standard in previous disentanglement works as well, metric evaluation requires full supervision even if training methods don't.
>
>
> (3) *LSBD is clearly defined, SBD is mentioned multiple times but never defined.*
>
> We will add the definition of SBD to the Appendix.
>
>
> (4) *The ∆VAE model is never described, nor its architecture. The reader is required to read another paper to understand how this model works*
>
> We will add details about the ∆VAE model both to the main text and to the Appendix. Specific information about the architecture will be included.
>
>
> ... to be continued (1/2)
>
>
> [1] Francesco Locatello, Stefan Bauer, Mario Lucic, Sylvain Gelly, Bernhard Sch ̈olkopf, and OlivierBachem.  Challenging common assumptions in the unsupervised learning of disentangled repre-sentations. ICML 2019

---

> > ### Author Response · Authors · 2020-11-21
> > **Author response (2/2)**
> >
> > (5)*"An easy-to-compute metric to quantify LSBD given certain assumptions (see Section 4), which acts as an upper bound to a more general metric (derived in Appendix C)." This more general metric is not described in the main text, and it is unclear what the underlying motivation is for this alternative metric (and reading the technical description in the appendix did not help me understand the motivation).*
> >
> > (6)*Enigmatic discussion points: "Our LSBD metric and method require a number of assumptions, as explained in Section 4. This limits the applicability of the metric and method, but also provides a clear direction to what needs to be done to obtain and quantify LSBD representations if these assumptions are relaxed.", "Moreover, our metric is in fact an upper bound to a more general metric (see Appendix C), which is however less straightforward to compute." What are the clear directions on what needs to be done? How to think of this more general metric?*
> >
> > For both concerns (5) and (6) we will explain the motivation for the general metric presented in the paper more clearly. As it has been pointed out (remark (1) of this reply), the metric that was practically used in the paper requires prior knowledge on the data and the structure of the group representations. The purpose of the general metric is to remove the requirements on the prior knowledge about the structure for the group representations.
> >
> > Recall that the general metric is given by the formula:
> >
> > $\mathcal{M}_{LSBD} =\inf_{\rho \in \mathcal{P}(G, Z)} \int_G \left\| \rho(g)^{-1} \cdot h(g \cdot x_1) - \int_G \rho(g')^{-1}\cdot h(g'\cdot x_1) d\nu(g') \right\|_{\rho, h, \mu}^2 d \nu(g)$
> >
> > with $\mathcal{P}(G, Z)$ denoting the set of disentangled group representations of $G$ in $Z$. The metric can be interpreted as the measurement of the average deviation from equivariance for the data encodings provided by the encoding function $h$ with respect to the best linearly disentangled group representation $\rho^\*\in\mathcal{P}(G, Z)$ that can be fitted on those encodings. However, it remains a challenge to find a method to identify the best candidate for $\rho^*$ within the search space of $\mathcal{P}(G, Z)$ that can be fitted. Thus, the practical implementation of such a metric is out of the scope for this paper.
> >
> > Regarding the words "a clear direction what needs to be done", we are referring to the relaxing of our (well-formalized) assumptions, and analyzing what parts of our metric computation are affected by this. We realize this formulation is a bit misleading, so we will update the text.
> >
> >
> > (7) *The LSBD assumptions at restated three times in the main text (p2,3,4), which is unnecessarily redundant.*
> >
> > Thank you for pointing this out, we will adjust the text to remove this redundancy.

---

### Official Review · AnonReviewer3 · 2020-11-01
**Quantifying and Learning Disentangled Representations with Limited Supervision**

**Rating:** 6
**Confidence:** 3

**Review:**

Summary: This paper follows on the work of Higgins et al. 2018 that used linear symmetry based disentangled (LSBD) representations where real-world transformations provide some structure in the data that can be leveraged. The main contribution of this paper is to provide a metric to measure the quality of disentanglement in the learned representation. The paper makes some assumptions on the samples in the dataset and assumes that there are some group actions that relate one data point to another. Under this assumption, the paper derives a simple and easy-to-compute disentanglement measure. Second, this paper shows a method that can work with partial supervision to learn disentangled representation. This is done by using some data with supervision on the underlying transformation between the data points, and another set of data points where no labeling information is given. The proposed method uses a diffusion variational autoencoder.

Pros:

1) This paper proposes a metric that can be used to quantify disentangled representations. Disentangled representation is one of the overloaded terms in the community, and there have been multiple interpretations without any clear evidence as to whether the learned representation is disentangled or not.
2) The problem formulation and setup has been nicely illustrated using observation and inference function modeling the different factors of variation in the data, and group actions that allow the transformation of one data point to another.


Cons:

1) My first concern is regarding the strong assumptions used in this paper. It is hard for me to think of real world data, where we can have simple group actions that map one data point to another. This restricts the use of the disentanglement measure proposed in this paper or real data.

2) While the paper clearly argues about the importance of having a measure for disentanglement, it does not tie this to the effectiveness of the downstream application. In practice, we are always interested in disentangled representations that also capture all the relevant information from the original data. While it might be possible for the inference function h to satisfy the equation h(g_n . x_1) = \rho (g_n). h(x_1)  as given in equation (2),  but this may not mean that the learned representation contains the information associated with the original data points.

3) The batch size used for the supervised data points is 2. This implies that there is a group action between only two samples. I am a bit concerned that this does not capture the generality of the proposed formulation proposed in section 5.

4) The main algorithm proposed in this paper is based on the diffusion variational autoencoder (Peez et al. 2020), and the method to handle unsupervised data is just by alternating the training between supervised and unsupervised data. There is not much analysis or ablation study on how much unsupervised data can be used.

5) No comparison is made with other disentanglement methods that use partial supervision in the form of set membership.

---

> ### Author Response · Authors · 2020-11-21
> **Author response**
>
> Dear reviewer, we would like to thank you for the time and effort spent in analyzing this paper and for the specific suggestions made to improve this work. We will address the concerns presented in the review in the same order as presented.
>
> Comments on the cons:
> 1. *My first concern is regarding the strong assumptions used in this paper. It is hard for me to think of real world data, where we can have simple group actions that map one data point to another. This restricts the use of the disentanglement measure proposed in this paper or real data.*
>
> We agree that our assumptions are limiting the applicability of the method & metric presented in this paper. However, we emphasize that the symmetry-based approach to disentanglement is still quite a new area of research, and we argue that strong formalism is needed before attempting to broaden the applicability of the method & metric we provide. We believe it's important to verify in more controllable settings first whether LSBD can indeed be achieved and evaluated. Moreover, note that previous (factor-based) disentanglement techniques are also typically tested on datasets with strong assumptions on the generative process (i.e. a small set of factors fully describes the data), but these assumptions haven't been as well formalized as they are now with the help of group theory. This lack of formalization has led to less well-understood methods & metrics, at the cost of losing reliability. Therefore we take a more cautious approach in our work, favoring formalism over immediate results on real-world data.
>
> We definitely think it is a vital next step to work towards the ability to deal with more general scenarios, but this is outside of the scope of this paper. To get there, we believe it's important to first formalize the assumptions we make in this paper, as a first step towards this goal. Future work then only needs to focus on how to relax these assumptions, such that our method and metric become applicable to a wider variety of real-world data.
>
>
> 2. *While the paper clearly argues about the importance of having a measure for disentanglement, it does not tie this to the effectiveness of the downstream application. In practice, we are always interested in disentangled representations that also capture all the relevant information from the original data. While it might be possible for the inference function h to satisfy the equation h(g_n . x_1) = \rho (g_n). h(x_1) as given in equation (2), but this may not mean that the learned representation contains the information associated with the original data points.*
>
> Under Linear Symmetry-Based Disentanglement, the assumption is that the group structure is in fact what describes the relevant information associated with the data. Thus, equivariance as in equation (2) is then directly linked to the informativeness of the latent representations. In particular, the linearity of group representations makes the requirement of LSBD even stronger than the notion of informativeness often used in other disentanglement works (not aiming for symmetry-based disentanglement).
>
>
> 3. *The batch size used for the supervised data points is 2. This implies that there is a group action between only two samples. I am a bit concerned that this does not capture the generality of the proposed formulation proposed in section 5.*
>
> Note that this batch size only refers to our training method, whereas the formulation in section 5 applies to the metric computation, which does take into account all transformations between data points (essentially it's having batch size N).
>
> Our results use batch size 2 to demonstrate our training methodology, since it is actually the most limited form of supervision. We show that even by showing only single "links" of transformations (note that these links are disconnected from other links), the method learns a more general description of the underlying transformations. Higher batch sizes would provide more information by connecting more data points together in terms of their transformations.
>
> We will revise our text to emphasize these points.
>
>
> ... to be continued (1/2)

---

> > ### Author Response · Authors · 2020-11-21
> > **Author response (2/2)**
> >
> > 4. *The main algorithm proposed in this paper is based on the diffusion variational autoencoder (Peez et al. 2020), and the method to handle unsupervised data is just by alternating the training between supervised and unsupervised data. There is not much analysis or ablation study on how much unsupervised data can be used.*
> >
> > We demonstrate our method on a given dataset, from which we select a certain number of unique data pairs for which we label the in-between transformation. This selection of pairs is done randomly, different for each experiment. We show results for various numbers of labelled pairs, to showcase the effect of the ratio between supervised and unsupervised data. In particular, our results include the case where only unsupervised data is used, as well as the case where all data points are involved in a supervised pair. Moreover, supervised data is used to optimise a loss function that is very similar to the unsupervised loss (the ELBO), with an additional term for the supervision. We believe there is not much else to show in terms of how much unsupervised data can be used.
> >
> >
> > 5. *No comparison is made with other disentanglement methods that use partial supervision in the form of set membership.*
> >
> > We emphasize that we focus on symmetry-based disentanglement (Higgins et al., 2018), whereas previous work uses a factor-based definition of disentanglement. These definitions have different implications, in particular LSBD requires certain properties from disentangled representations that will simply not arise in previous disentanglement methods. Therefore we think a comparison would be unfair (favoring our method for the LSBD metric). Vice versa it is similar; previous disentanglement metrics typically want a single latent dimension to encode a single generative factor, whereas we show that LSBD requires at least 2-dimensional subspaces for SO(2) group structures. Existing disentanglement metrics cannot deal with this, or would need to be adjusted.
> >
> > We do believe further study into the differences between factor-based and symmetry-based disentanglement is a very interesting and useful topic, but it is outside of the scope of this paper. At this point we believe a decent comparison with other methods cannot be made, and would therefore only be misleading. Instead our experimental results serve as a sanity check for our LSBD metric, and as an investigation into the effect of supervision.

---

### Decision · Program_Chairs · 2021-01-07
**Final Decision**

**Decision:**

Reject

**Comment:**

This paper proposes a new metric to measure symmetry-based disentanglement and uses this metric to optimize diffusion VAEs on a set of small, synthetic datasets. In general, reviewers found the theoretical framework introduced to be interesting and relevant, but there were a number of concerns regarding the empirical evaluation in the paper and the clarity of many of the claims, particularly wrt the need for strong supervision (pairs of data points with a known transformation between them) for both evaluating the metric and for training by regularizing the proposed metric. I'd encourage the authors to focus on the improvement points suggested by reviewers, most notably by improving the empirical evaluation by adding detailed ablations and comparisons (e.g., exploring the relative amount of supervision needed, comparisons to previous approaches) and clarity regarding the supervision required. As such, I recommend that it be rejected in its current form.